# Association between Soccer Participation and Liking or Being Proficient in It: A Survey Study of 38,258 Children and Adolescents in China

**DOI:** 10.3390/children10030562

**Published:** 2023-03-16

**Authors:** Yibo Gao, Xiang Pan, Huan Wang, Dongming Wu, Pengyu Deng, Lupei Jiang, Aoyu Zhang, Jin He, Yanfeng Zhang

**Affiliations:** 1China Institute of Sport Science, Beijing 100061, China; 2Graduate School of Health and Sports Science, Juntendo University, Inzai 270-1695, Japan

**Keywords:** influencing factors, level, regions, cross-sectional analysis

## Abstract

Soccer participation among children and adolescents is low in China. To achieve a coordinated development of soccer in all regions and to promote the physical health of children and adolescents, this study aims to identify the influencing factors regarding the participation of children and adolescents in soccer programs through a cross-sectional analysis of the “soccer population” of children and adolescents. A total of 38,258 children and adolescents aged 7–18 years were included in this study. In addition, the analysis was conducted by dividing the regions where the children and adolescents live into three parts according to socioeconomic status, and by incorporating five dimensions, including environment, family, school, community, and individual levels to find the influencing factors of children and adolescents’ participation in soccer. Chi-square test, Pearson’s correlation, and one-way logistic regression analyses were used. The results showed that the area (r = 0.487) and the average annual precipitation (r = −0.367) were associated with the participation of children and adolescents in soccer programs. Moreover, the percentage of children and adolescents who participated in soccer programs (24.5%) was higher than those who liked soccer or were proficient in it (14.4%). Meanwhile, parental encouragement and support (OR = 0.627; 95% CI, 1.102–3.179), as well as the accessibility (OR = 0.558; 95% CI, 1.418–2.155), availability (OR = 1.419; 95% CI, 1.179–1.707), and safety of sports facilities (OR = 0.316; 95% CI, 0.614–0.865), influence children and adolescents’ participation in soccer programs.

## 1. Introduction

Competitive sports are one of the best ways to showcase a country’s athletic achievements, and soccer has always been the most internationally popular representative competitive sport [1]. In reality, children and youth aged 7 to 18 years are most exposed to school soccer [2]. Expanding the soccer population, namely, increasing the spread and promotion of the sport among youth, is an issue on which countries are placing more importance [3]. According to the State General Administration of Sports, 6326 schools in China have established school soccer leagues with 191,800 registered players in 2015 [4]. Globally, however, the prevalence of soccer among Chinese children and adolescents is only 2% in European and American countries [5]. Moreover, China has introduced a series of policies to accelerate the popularization of soccer at the children and adolescents level, such as the “Overall Plan for the Reform and Development of Chinese Football”, in order to promote the development of Chinese soccer [6,7,8].

Previous studies have shown that mastery of soccer skills significantly improves metabolic, cardiovascular, and muscular function in children and adolescents, increases bone mineral density, self-confidence, and physical awareness, as well as increases the attractiveness of physical activity to them [9,10,11]. Adolescent soccer players have significantly improved flexibility, coordination, balance, speed, strength, and stamina [12,13]. In addition, participation in soccer helps in reducing the risk of injury and disease [14], increasing knowledge related to health, nutrition, and fitness activities [15], gaining a sense of well-being in life [16], and promoting physical health [17,18].

Studies have shown that the choice of sports to participate in is also influenced by the geographic location and natural environment [19,20]. Snow and ice sports in the Nordic countries and water sports in Australia and other countries have a high proportion of participants; therefore, the choice of sports for children and young people varies from region to region [21,22]. Second, factors from family and social environments, such as parents’ attitudes toward sports and the fitness environment in their neighborhood, may influence children and adolescents’ choice of sports programs [23,24]. Finally, the socioeconomic status of the family in terms of education, income, occupation, fitness equipment, and transportation costs incurred influence the choice of sports programs for children and adolescents [25]. However, current research has not yet conducted a comprehensive survey and analysis of the specific physical activities performed in different regions and by different individuals.

China is a vast country with significant differences in the natural and economic environment of each region. The eastern region has a better economy than the other two regions, in addition to a larger population and higher average annual precipitation. The western region covers a wider area, is sparsely populated, and has less average annual precipitation and more mountain ranges. At present, China’s soccer population density is less than 1.5%, compared with the soccer population density of 7–8% in the world’s leading soccer countries, and the soccer level of children and adolescents is at a relatively low level [26,27]. Therefore, the popularization of soccer in China still has a long road ahead. However, there is no research that solves this problem in China, since the reasons for the low participation rate in Chinese soccer are unknown.

In summary, many relevant factors influence the choice of sports programs for children and adolescents. Therefore, this study was conducted to find the factors that influence the participation of children and adolescents in soccer programs. By improving these factors, it is possible to encourage children and adolescents to benefit from soccer programs, to promote the development of physical health, to reduce the burden of families on the physical development of children and adolescents, and to provide recommendations to the state and society for the coordinated development of the “soccer population” of children and adolescents.

## 2. Materials and Methods

### 2.1. Participants

The study population consisted of children and adolescents aged 7–18 years in 31 provinces (autonomous regions and municipalities directly under the central government) in China. According to the proportional probability sampling method by size, the sample provinces (autonomous regions and municipalities directly under the central government) cover mainland China. A total of 10–20 counties (districts) are randomly selected in each province, 13 villages (neighborhood committees) are randomly selected in each county (districts), and survey respondents are randomly selected in each village (neighborhood committee). In particular, 5760 village (residential) committees in 471 counties (districts) were selected from all over the country, and a total of 38,258 children and adolescents were investigated using household entry and data uploading, with a final effective sample size of 35,653 and a recovery rate of 93.2%.

Survey participants were divided into three age groups: 7–9, 10–12, and 13–18 years. The number of participants in each age group and the number of people who participated in and liked or were proficient in the soccer program are shown in Table 1.

### 2.2. Division

China is a vast country, and the natural environment and socioeconomic status vary greatly from region to region; therefore, the regions of China were divided into three regions according to the needs of national economic development and the degree of economic development, the overall level of economic and technological development, and the geographical differences in natural and socioeconomic status [28] as detailed in Table 2.

### 2.3. Selection of Questions

The questionnaire was obtained from the Survey of National Fitness Activities questionnaire and passed the reliability analysis. Then, the questionnaire referred to the family, school, community, and individual levels. At the family level, four factors were selected concerning parents’ attitudes toward their children’s sports, living with parents, and sports opportunities in the family. At the school level, one factor was selected on whether the skills acquired in school will persist. Three factors were selected at the community level concerning the quality of facilities, convenience, and safety of the community environment. Three factors were selected at the individual level, for example, have you played soccer regularly in the past year, and have you been able to find friends or lose weight by playing soccer? The questionnaire was completed by the children and their parents or guardians for the 7–12 age group and by the children themselves for the 13–18 age group. The questions and options corresponding to each stage are listed in Table 3.

### 2.4. Proceduce

Prior to conducting the survey, the staff were first trained to familiarize themselves with the specific plan of this study. Information about respondents at each observation site was obtained from the local statistical office, and a random sample of those who met the requirements was selected. Respondents were contacted by telephone and asked for their consent before the survey began. Then, they signed a consent form following the receival of specific information about the process and purpose of the study. In addition, a written informed consent was obtained from the legal guardians of all minor participants. During the survey, respondents’ names were replaced with numbers, and personal information remained confidential throughout the survey process.

Questionnaires were completed after face-to-face interviews in specific communities, with each interview lasting approximately 1 h per participant, while quality control was performed through return visits. The quality control system was based on the Internet platform for collecting and managing information on fitness status 2020, and the main body consisted of the national research team (National Center for Physical Fitness Monitoring), the provincial research team (autonomous regions and municipalities directly under the central government), and the county (district) survey team, in which the village (neighborhood committee) of data collection was responsible for both quality control and supervision of the higher-level organizations and quality control of the data obtained from their work. All methods were carried out in accordance with relevant guidelines and regulations. The survey period was 9–11 January 2020, and full ethical approval was obtained from the China Institute of Sport Science, Beijing, China (CISS-2019-10-29).

### 2.5. Statistical Analyses

First, the data were obtained from the questionnaire and the questions were dichotomous variables. Second, in the comparative analysis between age groups in each division, the number of participants compared to “like me” or “dominate me” was expressed as a percentage and a Chi-square test was performed. In the process of the Chi-square test, we weighed the cases. In addition, characteristics by the percentage of participants and liking or proficiency were compared using Pearson’s Chi-square test (minimal expected value > 5) and Fisher’s Exact Chi-square test (minimal expected value ≤ 5) [29]. Third, in the process of Pearson correlation analysis, data of 31 provinces were tested for normal distribution, and correlation analysis was carried out by province (city, district) in each region of China and the whole country. In particular, Pearson correlation analysis was conducted between the number of participants and likeability or proficiency in each region and gross domestic product (GDP), provincial area, and average annual precipitation. Finally, a one-way logistic regression analysis was conducted to examine the factors associated with the influence of children and youth soccer. For logistic regression analysis, the dichotomous variables are assigned with “0 = no and 1 = yes”, and the multi-categorical variables are assigned with sequential numbering. The Benjamini-Hochberg method was used to reduce the false discovery rate (FDR) when performing multiple comparisons. After adjusting raw *p*-values with the Benjamini-Hochberg method to control the FDR level to 5%, unadjusted and adjusted logistic regression models were used to identify the four levels (family level, school level, community level, and individual level) associated with soccer participation. A subset of *p* < 0.05 was selected in the one-way logistic regression analysis and placed in the adjusted model. Then, the Spearman correlation test was applied to check the relationship between the subitems. If a moderate or high (*p* ≥ 0.30) correlation was found between two subitems, one of them was selected to be representative. Furthermore, the screened subscripts were jointly included in the regression equation for multifactor logistic regression analysis. The strength of the association was expressed as OR and 95% confidence interval (CI), and the difference was considered as statistically significant at *p* < 0.05 [30]. SPSS26.0 (IBM Corp., Armonk, NY, USA) was used for data analysis.

## 3. Results

The number of children and adolescents who participated in soccer sports programs accounted for 24.5% of the total number of respondents. Data from the different age groups on the number of people who participated in soccer sports programs showed the following characteristics: Age group 13–18 > age group 10–12 > age group 7–9, and the difference was statistically significant (*p* < 0.05). Children and adolescents who liked or were proficient in soccer sports programs accounted for 14.4% of the total number of people surveyed. The percentage was even lower, and it was consistent with the age group of participation in soccer sports programs when viewed in different age groups. Moreover, the difference was statistically significant (*p* < 0.05). In each age group, 2.3% (*p* < 0.05) more people participated in soccer than those who liked or were proficient in soccer in the 7–9 age group, 2.9% (*p* < 0.05) more in the 12–18 age group, and 4.9% (*p* < 0.05) more in the 13–18 age group. However, the number of those who participated in the soccer sports program was 10.1% higher in the 7–18 age group than those who liked or mastered the soccer sports program (*p* < 0.05).

### 3.1. Overview of the Distribution of the Three Major Divisions

In the national division, the percentage of people who participated in soccer was higher than the percentage of people who liked or were proficient in soccer in each region and each age group. In addition, the difference was statistically significant (*p* < 0.001). Moreover, the 10–12 age group was higher than the other age groups (*p* < 0.001).

For the eastern region, in terms of participation in soccer sports program, the percentage of people in the 10–12 age group was 3.7% (χ^2^ = 12.295, *p* < 0.001) and 3.3% (χ^2^ = 12.096, *p* = 0.001) higher than the 10–12 and 13–18 age groups, respectively. In terms of liking or proficiency in the sport of soccer, the number of people in the 10–12 age group was 1.7% (χ^2^ = 4.210, *p* = 0.040) and 3.7% (χ^2^ = 17.766, *p* < 0.001) higher than the 10–12 and 13–18 age groups, respectively.

For the central region, the characteristics of the age groups largely match those of the east. In terms of participation in soccer sports programs, the 10–12 age group has the highest percentage, 25.2%. However, there was no significance between all age groups in terms of liking or proficiency in soccer.

For the western region, in terms of participation in soccer sports programs, the 10–12 age group has the highest percentage, 27.2%. The difference was significant (χ^2^ = 30.432, *p* < 0.001) compared to the 7–9 and 13–18 age groups (χ^2^ = 11.266, *p* = 0.001). This is followed by a higher percentage in the 13–18 age group than the 7–9 age group (χ^2^ = 8.932, *p* = 0.003). In terms of liking or proficiency in soccer sports programs, the percentage in the 10–12 age group was 14.5%, which was 3.8% (χ^2^ = 18.077, *p* < 0.001) and 2.2% (χ^2^ = 19.042, *p* = 0.005) higher than the 7–9 and 10–12 age groups, respectively. See Table 4 for more details.

The percentage of children and adolescents who participated and liked or were proficient in soccer was calculated by division. In the three major sub-regions, the highest number of soccer participants was 38.2% in the eastern region and the lowest was 27.2% in the central region, while the highest percentage of those who liked or were proficient in soccer was 37.1% in the western region. See Figure 1 for details.

### 3.2. Uneven Distribution across Regions with a Strong Correlation with the Natural Environment of Each Region

For the national area, the people correlation coefficient of 0.487 for the region of the percentage of children and adolescents who liked or played soccer well and the provinces showed a positive correlation. The people correlation coefficient for the average annual precipitation is −0.367 (*p* < 0.05), which is negative. People correlation coefficients for participation in soccer programs and provincial GDP were 0.430 (*p* < 0.05) and 0.403 (*p* < 0.05) for the 7–9 and 10–12 age groups, respectively, showing a positive correlation. Participation and liking/skill of children and adolescents aged 13–18 years in soccer sports were positively correlated with provincial land area and negatively correlated with mean annual precipitation. Participation in soccer sports programs was positively correlated with province size, but not as strongly correlated with preference or ability for sports programs.

The people correlation coefficients between children and adolescents’ participation in soccer sports programs and liking or proficiency and GDP in the eastern region were 0.652 (*p* < 0.05) and 0.591 (*p* < 0.05), respectively, and the results showed a positive correlation. The people correlation coefficients for children and adolescents participation in soccer programs with province size and average annual precipitation were 0.804 (*p* < 0.01) and −0.717 (*p* < 0.05), respectively, while the correlation coefficients for preference or proficiency for soccer programs were 0.882 (*p* < 0.01) and −0.620 (*p* < 0.05), respectively. Participation and preference or proficiency for soccer programs were positively correlated with province size and negatively correlated with average annual precipitation. In addition, they were negatively correlated with mean annual precipitation.

The western region is primarily larger and more economically backward compared to the eastern and central regions. There is no correlation between the individual age groups and the relevant factors.

For each age group, the correlation between the 7–9 and 10–12 age groups and GDP in the eastern region remained consistent with the nation. Participation and liking or proficiency in soccer programs for children and adolescents aged 13–18 years in the central region were strongly positively correlated with province size. See Table 5 for details.

### 3.3. Correlation Factors Affecting Children and Adolescents’ Participation in Soccer

Due to the large difference between the number of individuals who were proficient in the sport of soccer and the number of individuals who participated in the above analysis, we performed a one-way logistic regression analysis to filter out significant factors (*p* < 0.05). The final factors corresponding to the four levels were identified: Family level (four factors), school level (one factor), community level (three factors), and individual level (three factors). The results of the multivariate logistic regression analysis show that the modified decidable coefficient is R^2^ = 0.297, indicating a good fit of the model. Moreover, a correlation analysis of the participation of children and adolescents aged 7–18 years in soccer sports programs was carried out.

Analysis of the relationship between participation in soccer sports programs and variables among children and adolescents aged 7–18 years was carried out through screening of four levels. At the family level, Y1 (OR = 1.756; 95% CI, 1.668–1.849; β = 0.563; *p* < 0.05), Y2 (OR = 1.194; 95% CI, 1.084–1.314; β = 0.177; *p* < 0.05), Y3 (OR = 0.106; 95% CI, 0.100–0.112; β = 2.247; *p* < 0.05), and Y4 (OR = 1.871; 95% CI, 1.102–3.179; β = 0.627; *p* < 0.05) all contribute to the children’s participation in soccer sports programs.

At the school level, Y5 (OR = 0.675; 95% CI, 0.626–0.727; β = 0.394; *p* < 0.05) all contribute to the children’s participation in soccer sports programs.

At the community level, Y6-1 (OR = 1.748; 95% CI, 1.418–2.155; β = 0.558; *p* < 0.05), Y6-2 (OR = 1.399; 95% CI, 1.136–1.723; β = 0.336; *p* < 0.05), Y7-1 (OR = 1.419; 95% CI, 1.179–1.707; β = 0.350; *p* < 0.05), Y7-2 (OR = 1.269; 95% CI, 1.057–1.523; β = 0.238; *p* < 0.05), and Y8 (OR = 0.729; 95% CI, 0.614–0.865; β = 0.316; *p* < 0.05) all contribute to the children’s participation in soccer sports programs.

At the individual level, Y9 (OR = 0.017; 95% CI, 0.015–0.018; β = 4.104; *p* < 0.05), Y10 (OR = 0.645; 95% CI, 0.532–0.782; β = 0.439; *p* < 0.05), and Y11 (OR = 0.634; 95% CI, 0.469–0.858; β = 0.455; *p* < 0.05) all contribute to the children’s participation in soccer sports programs. See Table 6 for details.

## 4. Discussion

The sports scene of Chinese children and adolescents basically comprises physical education classes in schools. In addition, the top five sports programs mainly include running, skipping rope, badminton, walking, and table tennis [31], and the number of participants in soccer sports programs is very small. However, soccer not only brings about healthy mental and physical development, but also fosters a sense of teamwork, promotes the development of motor skills, etc. [12,13,14,15,16,17,18]. In this study, a nationwide sample found that the number of children and adolescents who liked or were proficient in soccer was significantly lower than the number of people who participated in the sport. The larger the region and the higher the GDP, the more children and youth participate in soccer programs. Children and youth are more likely to participate in soccer if they live in a better socioeconomic status, if their parents encourage them to play the sport, and if they live in a better environment [6,7,8]. This study will not only provide guidance to the government and society to increase their “soccer population”, but will also enable all Chinese families to improve the factors that foster their children’s interest in soccer, in order that children and adolescents who are interested in soccer can participate in the sport, and communities in each region of China can improve the factors that promote the physical, psychological, and physiological benefits of soccer for children and adolescents in their communities [12,13,14,15,16,17,18].

Differences in children and adolescents’ participation and liking or proficiency in soccer sports programs and the proficiency of motor skills of children and adolescents both increase gradually with age and are inconsistent across all age groups [32]. In addition, as children and adolescents grow older, their cognitive abilities become more complex, and their fine motor development becomes more refined during subsequent growth and development [33,34]. Generally, the older you are, the easier it is to be proficient in a sport [35]. Therefore, the percentage of children and adolescents who liked or were proficient in soccer gradually increased with age. However, it is worth noting that it is unsuitable for all children and adolescents. Some children and adolescents may show negative emotions during exercise due to their late physical development, poor physical mobility, ability to learn motor skills, and motor perception, which may lead to an aversion to this program [36,37]. As a result, being proficient in a sport is based entirely on participation in the program and having a certain liking and love for the sport, in order to motivate children and adolescents to like or be proficient in the program. In this study, the larger Chi-square values and smaller *p*-values indicate a more significant variability, which is consistent with the significantly larger proportion of children and adolescents who participated than the percentage of those who liked or were proficient in soccer across all age groups. Conversion from participation to liking or ability in the soccer sports program shows wide variability across all regions and age groups.

Children and adolescents’ participation in soccer sports programs is influenced by many factors. Socioeconomic status is a general measure of a certain family class after understanding the relevant factors, such as educational level, income, and employment combined with economics and sociology [38]. Dumuid [39] and Kelishad et al. [40] showed a significant effect of family economic status on the physical activity of children and adolescents due to the high economic status in cities, higher family income, high parental education, the increasing availability of sports facilities and soccer clubs in more economically developed areas, and the fact that parents with higher economic status are more willing to raise their children [41]. The eastern regions are mostly coastal cities with more developed economies, and families in cities with higher socioeconomic status can be more supportive than rural families, where children and adolescents can learn more sports and join more sports activities [42], which can be attributed to some extent to participation in soccer sports programs. This is consistent with the result that the distribution of the “soccer population” of children and adolescents in China is higher on the eastern coast. Minuchin [43] elaborated on the dynamic systems view of the family environment, noting that families are composed of complex, multiple interdependent subsystems. In a follow-up study, it was found that children and adolescents’ choice of sports or active participation in physical activity was influenced by their parents’ physical activity behavior [44]. In addition, the study showed that the higher the level of parental sports participation, the higher the children tend to show motivation [45]. In a more in-depth study, Cong found that parental emotional support and encouragement had a significant effect on the increase in physical activity levels [46]. In conclusion, when parents have a positive effect on their children’s participation in sports, they can accelerate the spread of the “soccer population” among children and adolescents.

The natural environment is also a very important factor that influences the participation or preference and ability of children and adolescents in soccer. Research has proven that the relationship between participation in sports and the natural environment is two-sided. Specifically, the sport is influenced by the natural environment, while the natural environment is influenced by the sport [47]. Natural factors, in turn, show cross-correlations with the living environment, socioeconomic status, and sociocultural, and all these factors have a correlative effect on children and adolescents’ participation in soccer sports programs; therefore, the natural environment shows a correlation with participation in sports. More importantly, the soccer program requires soccer fields. Since soccer fields occupy an area of land, it is evident that the percentage of children and adolescents in the “soccer population” is higher in the eastern and central regions, both in economic terms and land area. The percentage of children and adolescents in the “soccer population” is negatively correlated with the average annual rainfall of each region due to the outdoor area.

Learning is at the core of children and adolescents’ development. Bauman [21] proposed that individuals can be influenced by five levels: Individual level, interpersonal level, environmental level, policy level, and global level. In addition, the socio-ecological model of physical activity is present throughout an individual’s life. At the same time, it should be noted that the interpersonal level and the environmental level play a dominant role in childhood and adolescence. Children and adolescents are exposed to a sports program in the school environment that is entirely teacher-led, and in which their classmates participate. In the larger campus environment, students participate in a sport that can motivate them. In addition, children and adolescents exchange sports tips or sports anecdotes from the school sports program. Therefore, interpersonal communication is an effective way to promote the participation of children and adolescents in this sport. It allows children and adolescents, through the influence of their environment and relationships, to become dominant in the sport [48] over time, from the beginning of their participation in soccer.

Sports facilities around the home, accessibility to sports venues, and the type and number of sports facilities that are suitable for children and adolescents influence children and adolescents’ participation in sports programs [49,50,51]. The greater the number of sports facilities and venues and the greater the accessibility to sports venues, the higher the number of sports options for children and adolescents. This suggests that the better the sports facilities in the community, the more children and adolescents can become a “soccer population”. In addition, the safety of the sports environment is an issue of concern to parents, such as sports venues located in local cities with heavy traffic and congested roads. In this case, sports danger substantially increased for children and adolescents; therefore, when participating in sports, the safety of sports venues is important [52]. This is consistent with the study results, and is an influential factor that affects the hindrance of children and adolescents when participating in sports programs.

The prerequisite for proficiency in soccer is participation in the sport [53]. Regular contact with sports can have a positive effect on the development of the “soccer population” over time. In addition, motivation in sports is one of the important regulators that motivate individuals to participate in sports [54]. The sport of soccer exhibits a high team-based level. By comparing children and adolescents who participated in soccer with those who did not, Nathan found that participation in the soccer program resulted in closer relationships with peers [55]. This is consistent with the results of this study, since children and adolescents seek more all-around physical development as they get older. The greater the sense of self-efficacy obtained in the sport, the more motivated the individual will be to participate in the sport [56]. By enhancing the self-efficacy of children and adolescents in soccer sports programs, this enabling factor accelerates its popularity.

## 5. Limitations

In this study, only four dimensions of the social-ecological theory were selected for comparative analysis, and relevant factors at the policy level were not included. Since the study population involved children and adolescents aged 7–9 years and the questionnaire was completed with the assistance of parents, the accurate measurement of their subjective attitudes was not a simple task. In addition, no comparative analysis (e.g., for gender) was performed. Another limitation was that although this study was a comprehensive survey covering mainland China, many regions in China were not analyzed. Moreover, the sample size of each region made it difficult to ensure the accuracy of the study results. Therefore, government policies will be incorporated into the further research process, the sample size of each province will be expanded to create two sample databases for boys and girls, and the screening of relevant factors will be improved by selecting representative provinces for further research.

## 6. Conclusions

In this study, the number of children and adolescents who liked or were proficient in soccer programs was significantly smaller than the number of people who participated in the sport. Moreover, the results demonstrated that the size of the area, the economic environment (GDP), and climatic factors could promote participation in soccer, as well as parental encouragement and support, and the availability, accessibility, and safety of sports facilities.

## Figures and Tables

**Figure 1 children-10-00562-f001:**
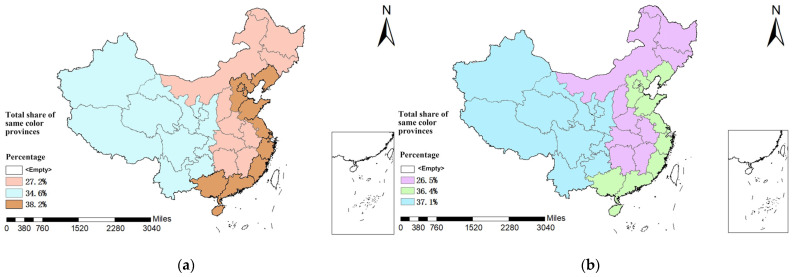
(**a**) Distribution of the three major regions in China and the percentage of children and adolescents who participated in soccer. (**b**) Distribution of the three major regions in China and the percentage of children and adolescents who liked or were proficient in soccer.

**Table 1 children-10-00562-t001:** Number of participants in each survey program and the number of people who participated in and liked or were proficient in soccer.

Age Group	Participated in Sports and Exercise Programs (People)	Participated in Soccer Sports Programs (People)	Liking or Proficiency in Physical Exercise Programs (People)	Liking or Proficiency in Soccer Sports Programs (People)
7–9 years	9290	2106	8569	1224
10–12 years	9488	2567	16,542	1456
13–18 years	16,875	4082	16,425	2194

**Table 2 children-10-00562-t002:** Regional division in China.

Three Regions	Provinces, Cities, Autonomous Regions and Municipalities Directly under the Central Government
Eastern Region	Beijing, Fujian, Guangdong, Guangxi, Hainan, Hebei, Jiangsu, Liaoning, Shandong, Shanghai, Tianjin, Zhejiang
Central Region	Anhui, Heilongjiang, Henan, Hubei, Hunan, Inner Mongolia, Jiangxi, Jilin, Shanxi
Western Region	Chongqing, Gansu, Guizhou, Ningxia, Qinghai, Shaanxi, Sichuan, Tibet, Xinjiang, Yunnan

**Table 3 children-10-00562-t003:** Options corresponding to the abstraction problem.

Level	Code	Question	Options
	X	Daily participation in sports is soccer	1 = Yes, 2 = No
Family	Y1	Have you purchased any sports-related products or services for your child in the past year?	1 = Yes, 2 = No
	Y2	Do the parents live with the child?	1 = Yes, 2 = No
	Y3	An issue that is often discussed at home is soccer	1 = Yes, 2 = No
	Y4	Do your parents often encourage you to participate in sports and activities?	1 = Yes, 2 = No
School	Y5	Will you keep doing the exercise program you learned in school?	1 = Yes, 2 = No
community	Y6	How convenient is it to go from your home to a nearby venue suitable for you and your friends to exercise?	1 = very convenient, 2 = more convenient, 3 = average, 4 = not very convenient, 5 = very inconvenient
	Y7	How satisfied are you with the sports facilities in your community or village?	1 = very satisfied, 2 = more satisfied, 3 = average, 4 = not very satisfied, 5 = very dissatisfied
	Y8	The degree of security in your home community or village	1 = very safe, 2 = more safe, 3 = average, 4 = not very safe, 5 = very unsafe
Individual	Y9	For the last year, the sport that has been consistently practiced is soccer	1 = Yes, 2 = No
	Y10	Physical exercise can help you find more friends	1 = absolutely agree, 2 = somehow agree, 3 = neutral, 4 = somehow disagree, 5 = absolutely disagree
	Y11	Physical exercise can help you lose weight	1 = absolutely agree, 2 = somehow agree, 3 = neutral, 4 = somehow disagree, 5 = absolutely disagree

**Table 4 children-10-00562-t004:** Percentage of “soccer population” by age group and three major regions.

	Age Group	Participation			Liking or Proficiency			Participation and Liking or Proficiency	
		%	Χ^2^	*p*	%	Χ^2^	*p*	Χ^2^	*p*
Eastern Region	7–9	24.7	12.295	<0.001	14.0	4.21	0.040	133.191	<0.001
	10–12	28.4			15.7			164.891	<0.001
	7–9	24.7	0.177	0.674	14.0	3.709	0.054		
	13–18	25.1			12.6			292.871	<0.001
	10–12	28.4	12.096	0.001	15.7	17.766	<0.001		
	13–18	25.1			12.6				
Central Region	7–9	21.1	11.901	0.001	12.3	1.463	0.226	71.658	<0.001
	10–12	25.2			13.4			119.796	<0.001
	7–9	21.1	3.325	0.068	12.3	0.009	0.924		
	13–18	23.0			12.4			194.418	<0.001
	10–12	25.2	4.588	0.032	13.4	1.716	0.19		
	13–18	23.0			12.4				
Western Region	7–9	21.5	30.432	<0.001	12.9	18.077	<0.001	80.919	<0.001
	10–12	27.5			16.7			107.965	<0.001
	7–9	21.5	8.932	0.003	12.9	4.421	0.036		
	13–18	24.3			14.5			186.419	<0.001
	10–12	27.5	11.266	0.001	16.7	7.836	0.005		
	13–18	24.3			14.5				
National	7–9	22.7	40.866	<0.001	13.2	19.042	<0.001	284.616	<0.001
	10–12	27.2			15.4			389.773	<0.001
	7–9	22.7	3.826	0.050	13.2	0.021	0.884		
	13–18	24.2			13.2			665.182	<0.001
	10–12	27.2	28.365	<0.001	15.4	23.629	<0.001		
	13–18	24.2			13.2				
7–18	Eastern Region	25.9	24.39	<0.001	13.9	7.494	0.006	587.22	<0.001
	Central Region	23.1			12.6			384.106	<0.001
	Eastern Region	25.9	7.662	0.006	13.9	3.271	0.071		
	Western Region	24.4			14.7			374.491	<0.001
	Central Region	23.1	5.263	0.022	12.6	19.457	<0.001		
	Western Region	24.4			17.7				
	National	24.6			13.8			374.491	<0.001

**Table 5 children-10-00562-t005:** Correlation analysis of the “soccer population” of children and adolescents with the natural environment.

		Aged 7–9		Aged 10–12		Aged 13–18		Aged 7–18	
		Participation	Liking or Proficiency	Participation	Liking or Proficiency	Participation	Liking or Proficiency	Participation	Liking or Proficiency
National	X	0.430 *	0.355 *	0.403 *	0.186	0.113	−0.088	0.314	0.138
	Y	0.165	0.258	0.156	0.353	0.472 **	0.638 *	0.315	0.487 *
	Z	−0.109	−0.171	−0.132	−0.333	−0.365 *	−0.449 *	−0.241	−0.367 *
Eastern Region	X	0.651 *	0.641 *	0.641 *	0.514	0.597	0.494	0.652 *	0.591 *
	Y	0.016	−0.017	0.154	0.069	0.266	0.253	0.159	0.100
	Z	0.347	0.049	0.250	−0.095	0.251	−0.071	0.296	−0.039
Central Region	X	−0.088	−0.055	−0.023	−0.159	−0.190	−0.330	−0.129	−0.237
	Y	0.545	0.639	0.492	0.795 *	0.804 **	0.882 **	0.691 *	0.850 **
	Z	−0.735 *	−0.694 *	−0.723 *	−0.755 *	−0.717 *	−0.620 *	−0.753 *	−0.709 *
Western Region	X	−0.013	−0.188	0.385	−0.132	0.288	0.052	0.246	0.086
	Y	0.380	0.466	0.099	0.409	0.338	0.589	0.326	0.543
	Z	−0.287	−0.139	−0.022	−0.061	−0.395	−0.525	−0.303	−0.461

* *p* < 0.05; ** *p* < 0.01; X: Total GDP (billion yuan); Y: Province area (million square kilometers); Z: Average annual precipitation (mm).

**Table 6 children-10-00562-t006:** Binary logistic regression analysis affecting the participation of children and adolescents aged 7–18 years in soccer sports programs.

Level	Variables	β	Wald	*p*-Value	OR	95% CI	
Family	Y1	0.563	457.197	0.000	1.756	1.668	1.849
	Y2	0.177	12.987	0.000	1.194	1.084	1.314
	Y3	2.247	5400.495	0.000	0.106	0.100	0.112
	Y4	0.627	05.371	0.020	1.871	1.102	3.179
School	Y5	0.394	105.310	0.000	0.675	0.626	0.727
Community	Y6-1	0.558	27.326	0.000	1.748	1.418	2.155
	Y6-2	0.336	9.990	0.002	1.399	1.136	1.723
	Y7-1	0.350	13.757	0.000	1.419	1.179	1.707
	Y7-2	0.238	6.502	0.011	1.269	1.057	1.523
	Y8	0.316	13.138	0.000	0.729	0.614	0.865
Individual	Y9	4.104	5781.784	0.000	0.017	0.015	0.018
	Y10	0.439	19.834	0.000	0.645	0.532	0.782
	Y11	0.455	8.705	0.003	0.634	0.469	0.858

## Data Availability

GDP, province area, and average annual precipitation can be found in the statistical yearbook of the National Bureau of Statistics of China. In addition, other data presented in this study are available on request from the corresponding author. The data are not publicly available due to privacy.

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
