# Peer review of "Association between Soccer Participation and Liking or Being Proficient in It: A Survey Study of 38,258 Children and Adolescents in China"

_children, 2023, doi:10.3390/children10030562_

Round 1

Reviewer 1 Report

Dear,

The research is interesting, the chosen statistical methods are logical, and results are interpreted in detail and well compared with similar available literature.

However, there are errors (technical) in nature that need to be corrected.

Please find my comments in a separate Word document.

Author Response

Dear,

We would like to thank the editors and reviewers for their valuable comments, which were very helpful in improving the quality of the manuscript. We have carefully studied these comments and have made some changes to the manuscript to the best of our ability. These changes will not affect the content and framework of the paper. I have responded to the reviewers' comments point by point and have highlighted these changes in the revised manuscript.  Note: The manuscript was revised in revision mode.
We sincerely hope that you are satisfied with our responses and revisions and that the manuscript is now ready to be accepted for publication. 
Looking forward to your reply.

Reviewer 2 Report

Dear authors

Regarding the present research titled: “Correlation Analysis Between Soccer Participation and Liking or Being Proficient in it: A Study with Children and Adolescents in China”

Despite the expressive sample size and the important topic, I noted several weaknesses in visualizations, background, results reports, and discussion descriptions In the present paper.

In this sense, have some suggestions, which, in my viewpoint, are necessary to reach the quality required by this journal.

 Follow my suggestions topic-by-topic:

Title

Replace correlation analysis per "Association", then, reduce the number of words in the title. In addition, I recommend adding "A Survey Stud" because it characterizes better the overall art

Keywords

Try not to repeat the same words present in the title in the keywords. In this sense, replace these keywords

Introduction

There are several mistakes in punctuation, for example, there are periods before the citations, while they must be after. Pay attention to these points in the text.

Lines 63-69: I think that, at this point, you should explain generally, about each China region, and this discrepancy between natural and economic environments. For example, which places are more or less favored economically?

Lines 71-78: Joint the texts of the two sentences, then, reorganize a new objective, shorter and more objective

Materials and Methods

There are punctuation and concordance mistakes in the text. Please, correct it

Division

Lines 98-100: This sentence is confusing, rearrange the text, please

Tables 2 and 3: Regarding the two tables inserted, spaces must be organized, and information is mixed and confused in the manuscript. Organize it, please

Statistical analyses

Please, the statistical analysis may present the citations of each procedure. In addition, you wrote about multiple logistic regression in the results, but not described it in the statistics topic, why did you not explain this procedure? If necessary, make it clear in the statistical topic. Moreover, look to organize better the text of statistical procedures

Results

Regarding the analysis of the proportion, why did you not add the correspondent effect sizes? Effect size is an important complement to the results. In this way, I advise you to perform this analysis for the analysis of the proportion.

Table 4: Please, replace the symbol “@” with a more adequate statistical symbol, there are other more used symbols.

Figure 1: The quality of the image should be improved also.

In the topic  “Uneven distribution across regions with a strong correlation with the natural environment of each region”: You report only coefficients of correlations, but, did not report p-values in the text, but, in the topic “Correlation factors affecting children and adolescents' participation in soccer” you reported confidence intervals, beta statistics values, and p-values, so, it is inconsistent between topics. I suggest that you must standardize it into two topics. Therefore, I feel the necessity to report the exact p-values in the text, equal are described in table 6, preferentially, the exact p-values must be reported always when cited.

Discussion

In the discussion, effects sizes (the meaningfulness of the correlations must be cited briefly), because it helps to comprehend the results in an applied mode. Furthermore, you did not discuss the benefits for health, provided by higher soccer participation, and like the sport, in this sense, I perceive the necessity of rearranging the discussion and valorizing these aspects.

No further comments.

Best regards;

Author Response

Dear,

We would like to thank the editors and reviewers for their valuable comments, which were very helpful in improving the quality of the manuscript. We have carefully studied these comments and have made some changes to the manuscript to the best of our ability. These changes will not affect the content and framework of the paper. I have responded to the reviewers' comments point by point and have highlighted these changes in the revised manuscript.  Note: The manuscript was revised in revision mode.
We sincerely hope that you are satisfied with our responses and revisions and that the manuscript is now ready to be accepted for publication. 
Looking forward to your reply!

Reviewer 3 Report

I have no problem with the science of the paper.  I have no problem with how the paper is written.  I have no problem with the measurement and statistical examination.  I have no problem with the findings and results. 

My problem and concern is on the merit of the research in general - more specifically what is the good of this research.  In the US, our IRB asks questions about the good that the participant gains from the study; the good that the discipline gains from the study; the good that the profession gains from the study; and the good that society gains from a study.

On Page 2, the aim is given which is about participation factors.  An IRB was approved, but the authors never address why this study was beneficial to the 35,653 people solicited, not the 93.2% who responded.  Nor is given the number of the children assenting or the good that was done from the study for the children, or the parents, or the discipline.  Perhaps the profession gained a benefit because more soccer problems would be developed for those who lead soccer programs... and I guess the state benefited to improve participation for more soccer participants which would indirectly help the state.  I recommend this be returned to the authors to consider the above.  Data is always good, but the data and the results should benefit the participants.... at least consider what you are doing in relation to IRB standards.  

I can not support this study until the authors describe the benefit for the participants and their families.  This is a study about human beings... the authors need to address the benefit to those human beings rather than the benefit to a coordinate program. 

Author Response

(The authors gave the same response as above.)

Reviewer 4 Report

The peer-reviewed manuscript is of interest to the readers of the journal. The introduction is informative - normative in parts - but overall successful. The description of the sample, questionnaire, variables, etc. is comprehensible. The statistical methods are adequate and purposeful for the research question. The results are presented logically and give a good impression. The discussion is rather a summary of the results and should be further elaborated. 

Missing is a discussion of the fact that different countries have different sports socialization and are subject to historical influences. This should be shown especially in these studies. Therefore, the following questions would still need to be answered:

Which other sports do the children and adolescents perform?

What is the time commitment in these other sports?

Are metropolitan regions not more inclined to "western" sports than more "rural" ones?

Doesn't the demand also correspond to the supply of soccer clubs?

The authors should still provide answers to these questions in the discussion.

Author Response

(The authors gave the same response as above.)

Round 2

Reviewer 2 Report

Dear authors,

I noted the significant improvements in your manuscript. I only have some more appointments in the pdf document regarding the missing writing and figures quality.

Additionally, I suggested you change your title from

“Association Between Soccer Participation and Liking or Being Proficient in it: A Survey Study with Children and Adolescents in China”

to

“Association Between Soccer Participation and Liking or Being Proficient in it: A Survey Study with 38,258 Children and Adolescents in China”, because it should be very attractive for readers.

Finally, I advise you to perform a deep English and text cohesion review and correction, as well as English connector usage along the manuscript.

Good job,

No further comments

Author Response

Dear,

We would like to thank the editors and reviewers again for their valuable comments, which will be another greater improvement to the quality of our manuscript. We have carefully studied the comments and made some changes to the manuscript to the best of our ability. These changes do not affect the content or framework of the paper. I have responded to the reviewers' comments line by line and have highlighted these changes in the revised manuscript. We sincerely hope that you are satisfied with our responses and revisions. And we hope that the manuscript is now ready to be accepted for publication.

Note:The first revision of the manuscript was revised using the revision mode, and the second revision has been highlighted in yellow.

Looking forward to your reply.

Reviewer 3 Report

The paper is much improved and now is truthful in what is being presented.  I am not a fan of this sort of work, but you met the requests.   I don't believe the paper is the greatest thing since sliced bread and is acting as a tool for the government, which can be a good thing or a very malevolent thing.  I believe it tends toward a beneficent study.  

Author Response

(The authors gave the same response as above.)
